# SCALING RELATIONSHIP ON LEARNING MATHEMATICAL REASONING WITH LARGE LANGUAGE MODELS

## ABSTRACT

Mathematical reasoning is a challenging task for large language models (LLMs), while the scaling relationship of it with respect to LLM capacity is under-explored. In this paper, we investigate how the pre-training loss, supervised data amount, and augmented data amount influence the reasoning performances of a supervised LLM. We find that pre-training loss is a better indicator of the model's performance than the model's parameter count. We apply supervised fine-tuning (SFT) with different amounts of supervised data and empirically find a log-linear relation between data amount and model performance, and we find better models improve less with enlarged supervised datasets. To augment more data samples for improving model performances without any human effort, we propose to apply Rejection sampling Fine-Tuning (RFT). RFT uses supervised models to generate and collect correct reasoning paths as augmented fine-tuning datasets. We find with augmented samples containing more distinct reasoning paths, RFT improves mathematical reasoning performance more for LLMs. We also find RFT brings more improvement for less performant LLMs. Furthermore, we combine rejection samples from multiple models which push LLaMA-7B to an accuracy of 49.3% on GSM8K which outperforms the supervised fine-tuning (SFT) accuracy of 35.9% significantly.

## 1 INTRODUCTION

Large language models (LLMs) (Anil et al., 2023; Touvron et al., 2023b; OpenAI, 2023) have shown considerable abilities in various math reasoning tasks (Saxton et al., 2019; Cobbe et al., 2021; Lightman et al., 2023). It is of interest to understand, predict, and improve an LLM's math reasoning ability based on different pre-trained LLMs and supervised datasets. With this knowledge, we can better decide the effort we put into improving the LLM or augmenting the dataset. Many recent works are focusing on using different prompts (Wei et al., 2022b; Yao et al., 2023) or ensembling / reranking multiple times of inferences (Cobbe et al., 2021; Uesato et al., 2022; Wang et al., 2023; Lightman et al., 2023) to improve models' reasoning performances. While in-context learning (ICL) and performing multiple inferences can improve performance, it is computationally expensive and not suitable for online deployment scenarios. Therefore, we focus on the performance of the supervised LLMs with inference only once which is a setting closer to online deployment.

To this end, we empirically investigate the scaling relationship of factors that influence the math reasoning abilities of a supervised LLM, including pre-training losses, the amount of supervised data, and the amount of augmented data. Firstly, we analyze the supervised fine-tuning (SFT) and ICL performance of LLMs. We observe that the pre-training loss is approximately negatively linear correlated to the SFT and ICL accuracy in a given interval which is a better performance indicator than pre-trained model sizes or pre-trained token counts. Secondly, we analyze the relationship between SFT and different amounts of supervised data. We observe that the model performance has a log-linear relation versus the supervised data amount while the increase diminishes with the better pre-trained model. Thirdly, we want to leverage the model itself to generate more supervised data to reinforce its reasoning ability and analyze the scaling relationship of the augmented data amount. We apply rejection sampling on SFT models to sample and select correct reasoning paths as augmented dataset (Uesato et al., 2022; Zhu et al., 2023). We use these augmented datasets to fine-tune base LLMs which would achieve better performances compared to SFT and we denote it as rejection sampling fine-tuning (RFT). We find the key factor influencing RFT performance is the

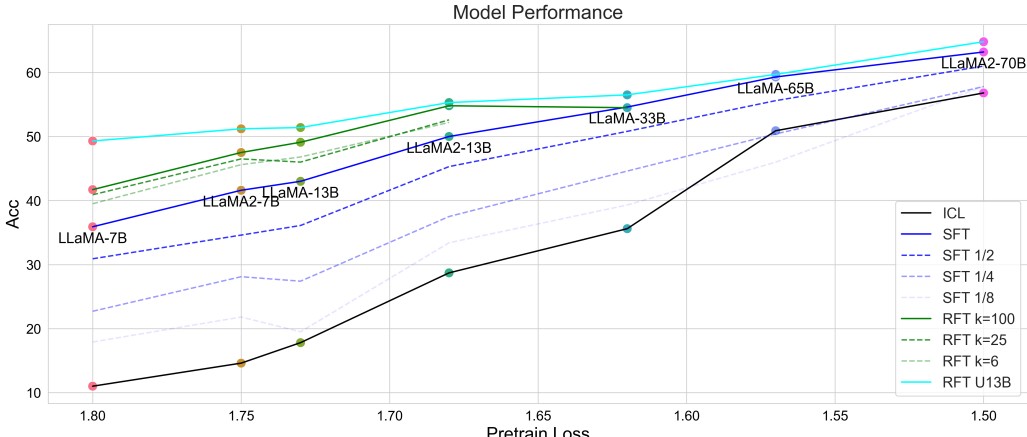

Figure 1: The key findings of scaling relationship on learning math reasoning ability with LLMs.

distinct reasoning path amount which can be increased by sampling more times or combing samples from multiple models. We apply RFT on several pre-trained LLMs and show larger improvement on less performant models. We discuss the reason RFT works is it provides multiple reasoning paths which makes LLMs have better reasoning generalization. We also discuss that RFT is much cheaper than pre-training in computational resources while training an LLM with lower pre-training loss is the fundamental solution.

The key findings of this paper are shown in Figure 1 and are summarized here:

- When the pre-training loss gets smaller , the model reasoning performances of SFT and ICL increase linearly within a range. The SFT performance improves slower than ICL.

- SFT improves in a log-linear manner with the increase of supervised data amount. The benefits of increasing data amount diminish as the pre-trained model gets better.

- The model performance for RFT improves as the distinct reasoning path amount increases. The RFT performance improves slower than SFT.

- The combination of rejection sampling samples from multiple models further enhances the RFT performance, resulting in an accuracy of 49.3 for LLaMA-7B (+13.4 compared to SFT), 50.3 for LLaMA2-7B (+8.7 compared to SFT), 52.1 for LLaMA-13B (+9.1 compared to SFT), and 55.4 for LLaMA2-13B (+5.4 compared to SFT).

## 2   RELATED WORKS

**Learning Math Reasoning with LLMs**   Recent research on LLMs has discovered the emergent ability to solve reasoning tasks beyond a certain model scale (Wei et al., 2022a). Such reasoning abilities in LLMs can be elicited by fine-tuning, few-shot prompting, or zero-shot prompting (Cobbe et al., 2021; Wei et al., 2021; Nye et al., 2021; Wei et al., 2022b; Kojima et al., 2022). A large amount of research focuses on the reasoning tasks of math word problems (MWP), and methods are evaluated on the benchmarks spanning different levels of MWPs (Koncel-Kedziorski et al. (2016); Patel et al. (2021); Lan et al. (2021); Cobbe et al. (2021); Jie et al. (2022); Yuan et al. (2023a); Fu et al. (2023a), *inter alia*). The core idea of improving the mathematical reasoning ability of LLMs is to aggregate various sampled reasoning paths during either fine-tuning or inference. Cobbe et al. (2021) trained and devised a reasoning path verifier to select the correct results during inference. Wang et al. (2023) proposed to sample various reasoning paths during inference and then derive the final result by majority voting on the answers or through verifiers (Li et al., 2023). Several works applied the idea of rejection sampling along with other techniques to filter the diverse sampled reasoning paths for fine-tuning data augmentation (Huang et al., 2022; Zelikman et al., 2022; Ni et al., 2023; Zhu et al., 2023). Rejection sampling is a simple yet effective fine-tuning augmentation technique and is also used for LLM alignment with human preference (Bai et al., 2022; Yuan et al., 2023b; Dong et al., 2023; Touvron et al., 2023b; Song et al., 2023). Uesato et al. (2022) explored to use of reinforcement learning methods for improving the mathematical reasoning abilities of

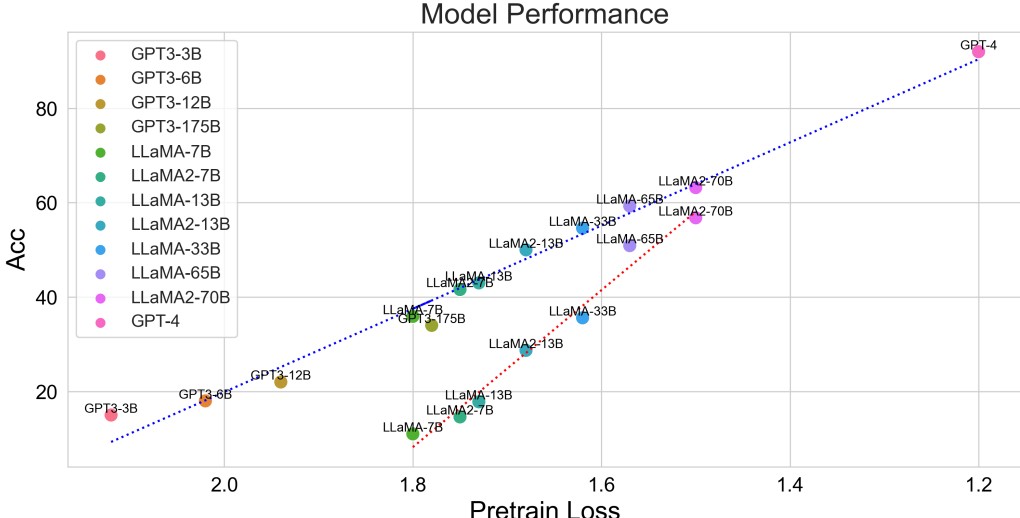

Figure 2: The performance of SFT (blue lines) and ICL (red lines) settings on GSM8K. GPT-4 states they use some part of the GSM8K data in pre-training, and suggest others consider its performance between SFT and ICL.

LLMs and they further discussed the difference between outcome-based and process-based reward modeling. Followed by Lightman et al. (2023), they collected large-scale process-based supervision signals through human annotation and verified that LLMs can benefit more from process-based reward modeling with human-annotated supervision than outcome-based reward modeling. There is also prior research that distilled the emergent reasoning ability of LLMs to small language models (Fu et al., 2023b; Shridhar et al., 2023). Compared to previous works (Zelikman et al., 2022; Uesato et al., 2022; Zhu et al., 2023; Ni et al., 2023), we are using a simpler way of generating augmented samples without any trained process-level reward models and we are focusing on researching the scaling relationship between LLMs and math reasoning ability.

**Scaling Laws of Large Language Models** It is important to understand and predict the performance gain as the language model scales up. Kaplan et al. (2020) first investigated and derived a predictable relationship on how the number of model parameters and data sizes contribute to the loss over many orders of magnitudes. Hoffmann et al. (2022) refined the scaling laws in (Kaplan et al., 2020) and found the scaling laws for computation-optimal training. Muennighoff et al. (2023) explored and extended the scaling laws under a data-constrained scenario. Besides investigating the scaling performance for pre-training, Gao et al. (2022) discussed the scaling laws for overparameterized reward models for alignment with human preference, and Hernandez et al. (2021) developed scaling laws for transferring performance from pre-trained models to downstream tasks. Henighan et al. (2020); Caballero et al. (2022) investigated scaling laws of math problems. In this paper, we are investigating the scaling relationships of large language models on learning math word problems with pre-training losses, supervised data amount, and augmented data amount.

## 3 THE FACTORS OF MATH REASONING ABILITY IN SUPERVISED LLM

The target of this paper is to try to understand the performances of supervised LLMs in math reasoning. We expect a pre-trained LLM $\rho$ to learn reasoning ability from a supervised reasoning dataset $\mathcal{D}$. The dataset is defined by $\mathcal{D} = \{q_i, r_i, a_i\}_i$, where $q$ is a question, $r$ is a chain-of-thought reasoning path, and $a$ is a numerical answer. We perform supervised fine-tuning on dataset $\mathcal{D}$ to obtain an SFT model $\pi$. We use $\pi$ to generate reasoning paths and answers in the test set by greedy decoding and report the accuracy (i.e. $acc$ or maj1@1) as our metric here.

### 3.1 MODEL ACCURACY VS. PRE-TRAINING LOSS

Previous works state that the larger LLM shows better reasoning ability across the same series of models (Brown et al., 2020; Chowdhery et al., 2022; Touvron et al., 2023a;b), and we find LLaMA

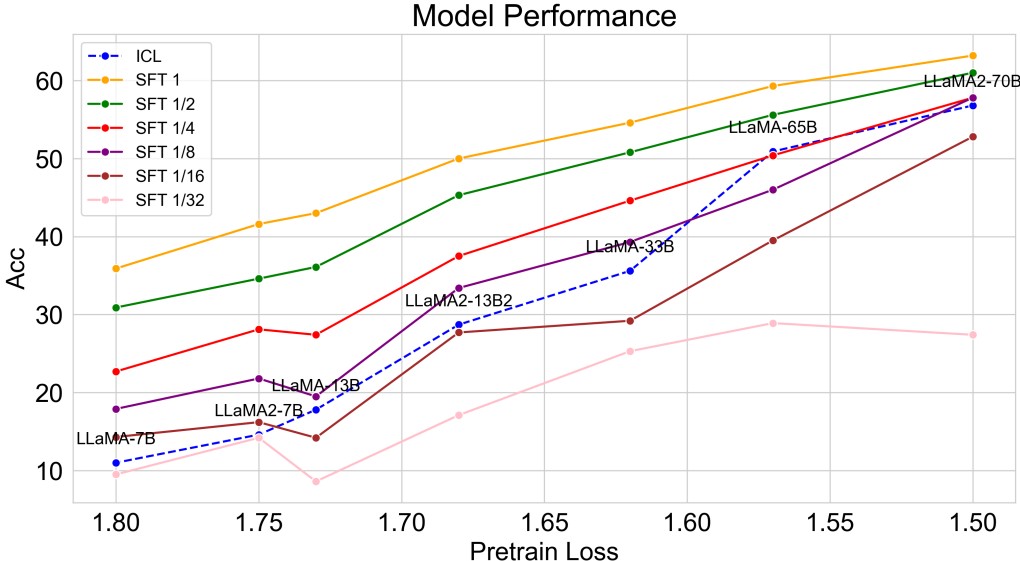

Figure 3: The performance of SFT with different amounts of supervised data on GSM8K.

outperforms GPT-3 which shows the model parameter counts should not be the only indicator of reasoning ability. While LLMs have different architectures, model parameters, and pre-training token numbers, we find the pre-training loss is a stable performance indicator of the math reasoning ability and we use it to represent the model instead of using their model parameters and pre-training token numbers.

We analyze the SFT and ICL (8-shot) performance of GPT-3 (Brown et al., 2020), LLaMA (Touvron et al., 2023a), LLaMA2 (Touvron et al., 2023b), and GPT-4 (OpenAI, 2023). The pre-training losses of these models are observed in their paper, we should notice that pre-training losses correspond to different pre-training datasets and different tokenizers which means they could not be compared strictly (and we cannot use it to do any sort of regression directly) while the tendency among these losses is still enlightening. We use the results of GPT-3 fine-tuning from (Cobbe et al., 2021) and we fine-tune LLaMA and LLaMA2 on the GSM8K training set (detailed in Appendix A.1). For in-context learning, we use the results from LLaMA (Touvron et al., 2023a) and LLaMA2 (Touvron et al., 2023b) paper.

In Figure 2, we can find that:

- The pre-training losses are approximately negatively linear correlated to the SFT and ICL accuracy during the given pre-training loss interval.

- SFT outperforms ICL consistently, while the improvements diminish when the pre-training loss is lower.

The linear relation of SFT and ICL accuracy may only work in the given interval. The reasons are (1) the slope of ICL is steeper than SFT, while the SFT performance should be greater than ICL performance; (2) the accuracy can not bigger than 1 or smaller than 0. It should be using $-\log(acc)$ instead of $acc$ as the dependent variable theoretically while we find an apparent linear relationship among pre-training loss and $acc$ and use $acc$ as the dependent variable. LLaMA-2 7B(13B) is a better counterpart of LLaMA 7B(13B) with a lower per-taining loss. Without changing the parameter count, ICL and SFT performance both improve significantly. From the observations, one effective way to improve reasoning ability is to train a better base model with lower pre-training loss. The models with lower pre-training loss improve less from the fine-tuning which may be due to the models having already obtained more reasoning abilities during pre-training and the supervised data can provide less signal to supervise them.

## 3.2 Model Accuracy vs. Supervised Data Count

Supervised fine-tuning does improve LLMs' reasoning ability, we want to know how the supervised data amount influences the model's improvement. We fine-tune LLaMA and LLaMA2 with $\{1, 1/2, 1/4, 1/8, 1/16, 1/32\}$ amount of the training set from GSM8K (detailed in Appendix A.2). We want to use this experiment to extrapolate the model performances if we have more supervised data. In Figure 3, we plot the results of training with different amounts of supervised data. From this figure, we can observe that:

- The model performance has a log-linear relation versus data amount. When the data amount doubles, the performance increases by a unit.
- Better model needs more amount of data to outperform its ICL performance.
- Better model benefits less when supervised data amount doubles.

The log-linear relation is stable during $\{1, 1/2, 1/4, 1/8\}$ amount of the training data. From the observation, it is straightforward to enlarge the training dataset to improve the performance, especially for worse models. For better models, it benefits less which echoes that better models have learned more reasoning ability during pre-training.

## 3.3 Model Accuracy vs. Augmented Data Count

Increasing the amount of math reasoning labeled data is difficult, especially for proposing a new question. It is easy for a well-educated student to solve hundreds of math word problems per day, but it is very hard to come up with diverse and educational math problems. So our direction changes to augment new data using existing resources. We have tried augmenting new queries (detailed in Appendix F.1) and augmenting revisions (detailed in Appendix F.2). These approaches have none to marginal improvements compared to SFT. We find a simplified version of rejection sampling (Zhu et al., 2023) is a naive and effective way to augment new reasoning paths and can improve the model performance. We find the key factor influencing fine-tuning on rejection sampling (RFT) augmented data is the distinct reasoning path amount. Combining rejection sampling samples from multiple models, we can further fine-tune a LLaMA-7B model to an accuracy of 49.3 (compared with SFT 35.9) and a LLaMA-13B model to an accuracy of 52.1 (compared with SFT 43.0).

**Rejection Sampling Fine-tuning** The SFT model $\pi$ obtains the ability to perform zero-shot chain-of-thought reasoning, and we use $\pi$ to generate more correct reasoning paths $r_{ij}$ to supply the training dataset. For each $q_i$, we generate $k$ candidate reasoning paths and answers $r, a$ with a temperature of 0.7 following (Cobbe et al., 2021). We first filter out reasoning paths with wrong answers $a \neq a_i$ or wrong calculations based on Python evaluation. Each reasoning path contains a list of equations $e_j$, and we select one reasoning path $r_{ij}$ for each distinct equation list as the augmented data and remove other reasoning paths with the same list of equations to deduplicate similar reasoning paths. Different order of elements (e.g. $3 + 4 = 7$ and $4 + 3 = 7$) or different order of equations (e.g. $1+2 = 3, 3+4 = 7$ and $1+4 = 5, 2+5 = 7$) are considered different. It is helpful for models to know these orders can be exchanged and is hard for models to learn this with only one reasoning path for each problem. We define $\mathcal{D}'_\pi = \mathcal{D} \cup \{q_i, r_{ij}, a_i\}_{i,j}$ as the augmented dataset. We fine-tune $\mathcal{D}'$ on pre-trained LLM $\rho$ to $\pi_{\text{RFT}}$ as RFT, and we detail how we apply RFT in Appendix A.3. We list the results of RFT with sampling $k = 100$ candidate reasoning paths on LLaMA and LLaMA-2 in Table 1. For ICL, SFT, and RFT, we list the maj1@1 (accuracy) and maj1@100 (sample 100 times and calculate accuracy based on majority voting) as metrics.

In the case of 7B and 13B models, RFT yields an approximate increase of 5 to 6 points in maj1@1 and about 4 points increase in maj1@100. For 33B models, RFT does not improve performance compared to SFT. The main reason comes from the augmented samples from rejection sampling. We can find that better models generate more correct reasoning paths per question. For LLaMA-33B-SFT, it can generate an average of 88.7 correct paths per question. However, it overfits the training set and has difficulty generating more diverse paths on the training set questions. Rejection sampling with 33B is very time-consuming and we do not conduct a temperate grid search, we have tried using a larger temperate 1.0 for decoding LLaMA-33B-SFT models, it generates 82.4 correct paths and 4.77 distinct paths per question which

| Setting | 7B | 7B-2 | 13B | 13B-2 | 33B |
|---|---|---|---|---|---|
| Pretrain loss | 1.8 | 1.75 | 1.73 | 1.68 | 1.62 |
| ICL | 11.0/18.1 | 14.6/- | 17.8/29.3 | 28.7/- | 35.6/53.1 |
| SFT | 35.9/48.7 | 41.6/55.4 | 43.0/55.2 | 50.0/61.7 | **54.6/-** |
| RFT $k = 100$ | **41.7/52.7** | **47.5/58.7** | **49.1/59.9** | **54.8/65.4** | 54.5/- |
| Correct paths per question | 53.3 | 60.8 | 62.5 | 71.6 | 88.7 |
| Distinct paths per question | 5.25 | 5.19 | 5.26 | 5.29 | 2.78 |

Table 1: The performance of RFT with $k = 100$ on GSM8K compared with SFT and ICL. Distinct path amount means distinct equation list amount here.

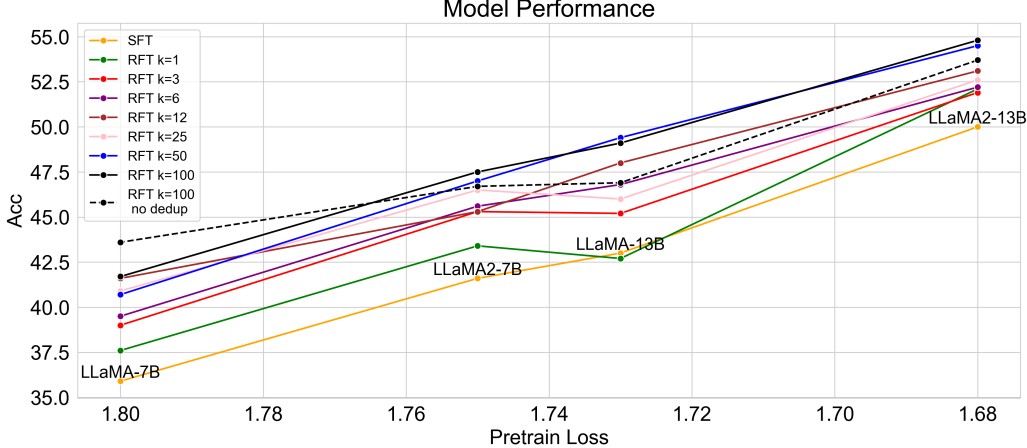

Figure 4: The performance of RFT with different amounts of sampling count $k$ on GSM8K.

is more diverse than using the temperate of 0.7 but still less diverse than 7B and 13B models. We admit there should be a temperate (or generation config) that can produce more distinct paths and generate good results for RFT in 33B and even larger models while it does need more computation resources for inference compared to sampling using 7B and 13B models. We will show we can use 7B and 13B models only for rejection sampling to improve the 33B model.

**Model Accuracy vs Rejection Sampling Data Count** To understand the performance of RFT, we vary $k$ among $1, 3, 6, 12, 25, 50, 100$ and apply RFT. We also have another setting of $k = 100$ while not removing any reasoning paths denoted as *no dedup*. We list the RFT results with different $k$ on Figure 4. Comparing using RFT $k = 100$ to *no dedup*, the performance is similar and shows that it is better to estimate RFT performance based on distinct reasoning path amount instead of RFT augmented sample counts. Furthermore, using deduplication has better performances for 3 of 4 models and needs much less training time.

| $k$ | 7B | 7B-2 | 13B | 13B-2 | 33B |
|---|---|---|---|---|---|
| 1 | 1.17 | 1.19 | 1.15 | 1.18 | 1.06 |
| 3 | 1.44 | 1.47 | 1.41 | 1.45 | 1.16 |
| 6 | 1.74 | 1.78 | 1.69 | 1.76 | 1.28 |
| 12 | 2.20 | 2.23 | 2.11 | 2.21 | 1.46 |
| 25 | 2.93 | 2.93 | 2.88 | 2.94 | 1.77 |
| 50 | 3.94 | 3.91 | 3.90 | 3.94 | 2.19 |
| 100 | 5.25 | 5.19 | 5.26 | 5.29 | 2.78 |
| $k_{\mathcal{D}'_{\text{U13B}}} = 400$, paths per question = 12.84 | | | | | |
| $k_{\mathcal{D}'_{\text{U33B}}} = 500$, paths per question = 13.65 | | | | | |

Table 2: Different reasoning paths per question generated by different SFT models with different $k$.

When using $k = 3$, RFT outperforms SFT by 2 points stably. For most data points, using larger $k$ leads to better performances. However, the merits of RFT are decreasing when doubling $k$. We calculate different paths per question for different $k$ in Table 2. We can see that the amount of different reasoning paths is not growing quickly along $k$ growing. In Figure 3, we know doubling training samples can have a linear performance improvement. Doubling reasoning paths should improve less than doubling training samples since obtaining different reasoning paths does not obtain any new questions. Therefore, doubling $k$ leads to diminished performance improvements.

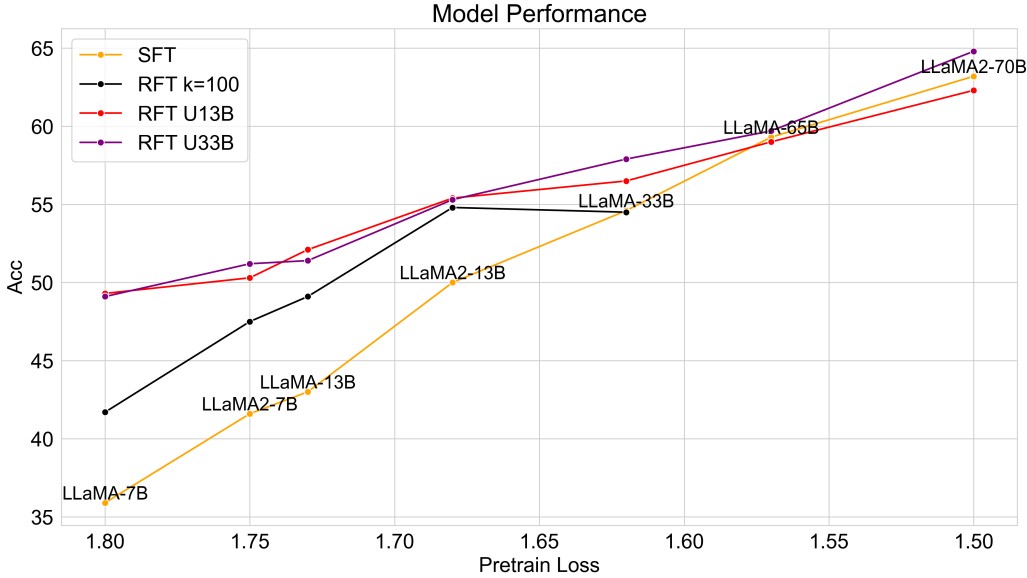

Figure 5: The performance of RFT with rejection sampling samples from multiple models.

**Combining rejection sampling samples from multiple models** The experiment results above demonstrate performance boosts in mathematical reasoning, benefitting from rejection sampling. Through case studies in 4, we show that rejection sampling can augment training data with reasoning paths of diverse calculation processes. However, the reasoning paths sampled from one single SFT model can be logically non-diverse. Therefore, we expect to further improve the mathematical reasoning performance by leveraging rejection sampled reasoning paths aggregated from different models. We denote two final datasets as $\mathcal{D}'_{U13B}$ and $\mathcal{D}'_{U33B}$, which are aggregated from rejection sampling different models $\mathcal{D}'_{U13B} = \mathcal{D}'_{7B} \oplus \mathcal{D}'_{7B2} \oplus \mathcal{D}'_{13B} \oplus \mathcal{D}'_{13B2}$ and $\mathcal{D}'_{U33B} = \mathcal{D}'_{U13B} \oplus \mathcal{D}'_{33B}$, where U means models under a certain size, 7B/13B/33B means LLaMA-7B/13B/33B and 7B2/13B2 means LLaMA2-7B/13B. $\oplus$ means an aggregation process in which all the reasoning paths from different sets

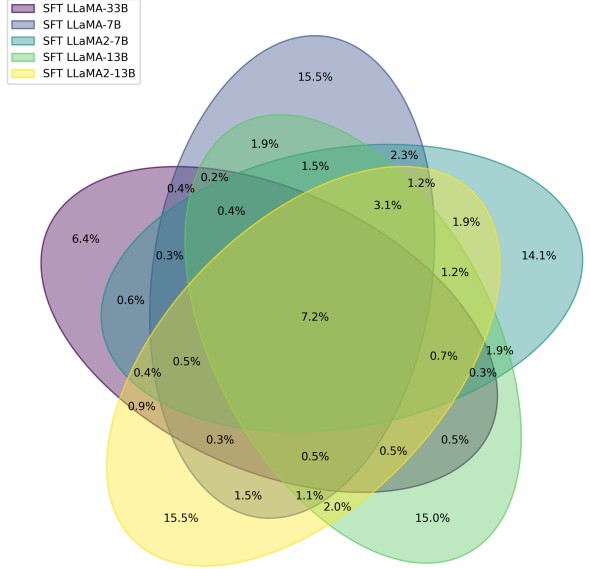

Figure 6: The Venn diagram of the proportions of the reasoning calculation paths that each model provides to $\mathcal{D}'_{U33B}$. For example, 15.5% (in the yellow part) of the reasoning calculation paths in $\mathcal{D}'_{U33B}$ can only be exclusively found in the rejection sampling results from LLaMA2-13B-SFT.

are first combined and then Algorithm 1 is applied to deduplicate the reasoning paths with the same calculation process regarding the equation forms and orders.

We can see, through the results visualized in Figure 5, that using the aggregated dataset $\mathcal{D}'_{U13B}$ and $\mathcal{D}'_{U33B}$ can lead to uniformly better performance than fine-tuning with datasets from a single model across different model sizes. RFT on these two augmented datasets $\mathcal{D}'_{U13B}$ and $\mathcal{D}'_{U33B}$ decreases the performance gaps among the same size models in SFT and RFT $k = 100$ which mean the combined

| Base Model | Training | maj1@1 | maj1@K* |
|---|---|---|---|
| **Proprietary LLMs** | | | |
| GPT-4 (OpenAI, 2023) | 5-shot ICL | 92.0 | - |
| GPT-3-175B (Brown et al., 2020) | SFT | 34.0 | - |
| PaLM2 (Anil et al., 2023) | 8-shot ICL | 80.7 | 91.0@K=40 |
| PaLM-540B (Chowdhery et al., 2022) | 8-shot ICL | 56.5 | 74.4@K=40 |
| Chinchilla-70B (Uesato et al., 2022) | 5-shot ICL | 43.7 | 58.6@K=96 |
| Chinchilla-70B | SFT | 58.9 | 77.7@K=96 |
| **Open-sourced LLMs** | | | |
| GPT-Neo-2.7B (Black et al., 2021) | FCS + PCS (Ni et al., 2023) | 19.5 | 41.4 |
| GPT-J-6B (Wang & Komatsuzaki, 2021) | CoRE (Zhu et al., 2023) | 34.9 | 63.2@K=40 |
| ChatGLM2-6B (Zeng et al., 2022) | 8-shot ICL | 32.4 | - |
| ChatGLM2-12B | 8-shot ICL | 40.9 | - |
| InternLM-7B (Team, 2023) | 4-shot ICL | 31.2 | - |
| LLaMA-7B | SFT | 35.9 | 48.7 |
| **Our RFT on open-sourced LLMs** | | | |
| LLaMA-7B | RFT-U13B | 49.3 | 61.8 |
| LLaMA2-7B | RFT-U13B | 50.3 | 65.6 |
| LLaMA-13B | RFT-U13B | 52.1 | 66.2 |
| LLaMA2-13B | RFT-U13B | 55.4 | 69.1 |

Table 3: Compare GSM8K results with other baselines. RFT-U13B means models fine-tuned on $\mathcal{D}'_{\text{U13B}}$. FCS and PCS represent fully-correct solutions and partially-correct solutions respectively. *K=100 if not specified.

augmented datasets provide enough reasoning supervision to fulfill the pre-training gap. We can assume with sufficient supervised data amounts, the performance indicator should be the model size but not the pre-training losses. We have stated that it is expensive to apply RFT $k = 100$ on 33B models and it needs a temperate grid search to achieve an improvement compared to SFT. However, fine-tuning on $\mathcal{D}'_{\text{U13B}}$ has a similar rejection sampling computational cost compared with sampling 100 times on 33B and achieves better performance.

Another phenomenon is including $\mathcal{D}'_{33B}$ in aggregation barely influences the performance. To give a more comprehensive analysis of the results, we calculate the average reasoning path number per question in Table 2 and depict a Venn diagram to visualize the source of different reasoning paths shown in Figure 6. In Table 2, the average reasoning path numbers of $\mathcal{D}'_{\text{U13B}}$ and $\mathcal{D}'_{\text{U33B}}$ surpass those of a single model by large amounts, while $\mathcal{D}'_{\text{U33B}}$ only have slightly more reasoning paths than $\mathcal{D}'_{\text{U13B}}$ by 0.81. In the meanwhile, as shown in Figure 6, the models under and including the size of 13B can contribute unique reasoning paths of similar proportion in $\mathcal{D}'_{\text{U33B}}$ around 15%. However, only 6.5% of the reasoning paths can be exclusively acquired from the LLaMA-33B-SFT model. This shows that the SFT model of 33B can provide limited reasoning diversity when sampling the training questions. This finding is consistent with the results above in Table 1, indicating the 33B model (and possibly 65B and 70B models) can well memorize the human-annotated reasoning paths. For 65B models, we find using $\mathcal{D}'_{\text{U13B}}$ does not improve the performance compared to SFT. The reason can be better models benefit less from the supervised sample amounts while it has learned more reasoning ability during pre-training.

Overall, we can come to the conclusion that (1) RFT improves the mathematical reasoning performance of (worse) LLMs through diverse reasoning paths from rejection sampling of the SFT models, and aggregating more diverse reasoning paths can improve the performance further. (2) Different SFT models can contribute reasoning paths with different calculation processes from rejection sampling, leading to more diverse training data for RFT, and LLMs of larger parameter sizes may degrade in generating diversified reasoning paths as a result of overfitting the training questions. There may be a generation config or training config for large enough LMs not to overfit on the training dataset while it is not trivial to find them.

**Comparing to other baselines** We compare our RFT results of training on $\mathcal{D}'_{\text{U13B}}$ to several baselines and the results are detailed in Table 3. Although LLaMA and LLaMA2 are top-tier open-sourced LLMs , their mathematical reasoning performances still lag behind the current proprietary LLMs which are of larger parameter scales, such as GPT-4 and PaLM2. Compared to results on open-resourced models, our results on LLaMA present better performance than two recent state-of-the-art reasoning augmentation methods. Our RFT method is simpler compared to CoRE, since

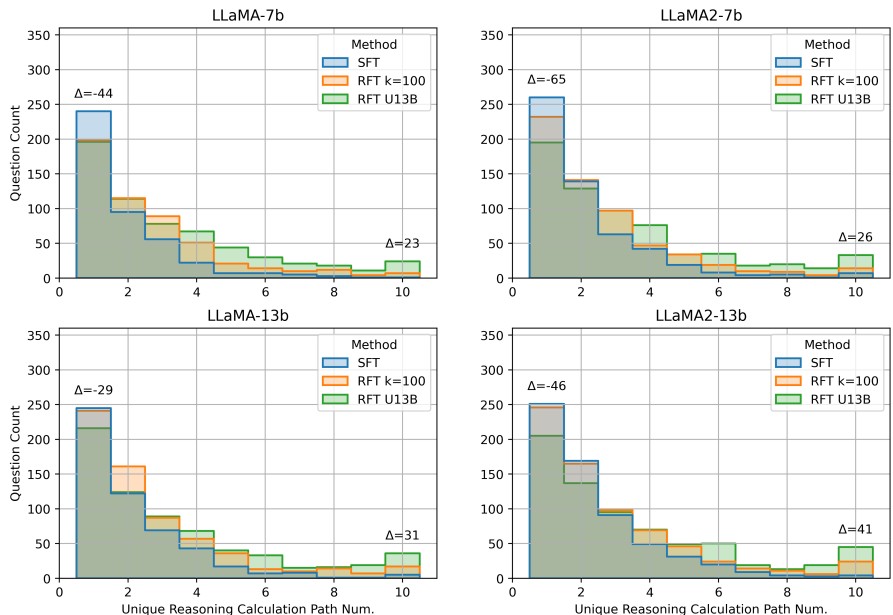

Figure 7: The histograms of question numbers solved with different numbers of unique reasoning calculation paths. We show the difference in question counts between SFT and RFT U13B in two cases where the numbers of unique reasoning calculation paths are 1 or more than 10.

RFT does not require training verifier models and decoding with Monte Carlo Tree Search (MCTS). Compared to other open-sourced aligned language models, we can find that 7B models struggle at a level of 35 scores which are very similar to SFT performances of LLaMA-7B. We guess they use GSM8K during their pre-training phase following (OpenAI, 2023) or human alignment fine-tuning phase following (Qingyi et al., 2023). Using our augmented dataset $\mathcal{D}'_{U13B}$ to replace the original GSM8K can significantly boost their 7B models' performances.

## 4 DISCUSSION

In the aforementioned analysis of RFT training data, we observe that rejection sampling can augment the training question with diverse reasoning calculation paths. In this section, we investigate whether RFT models can learn to generate different reasoning paths to reach the correct answers. We fine-tune LLaMA and LLaMA2 of 7B and 13B on $\mathcal{D}'_{U13B}$. During inference, we sample 100 different reasoning paths from each trained model for each test set question with a temperature of 0.7. For each question, we compute the number of different calculation processes presented in 100 sampled reasoning paths that lead to the correct answer and draw histograms with respect to test set questions. SFT and RFT models on self-sampled datasets (RFT k=100) are included for comparison.

As shown in Figure 7, the models trained by RFT on $\mathcal{D}'_{U13B}$ exhibit more question counts than the models trained by RFT k=100 and SFT on the larger numbers of unique calculation processes. There are more question counts for SFT models where all the sampled reasoning paths only correspond to one single calculation process and SFT models can barely generate more than 8 different calculation processes for a question. This analysis demonstrates that diverse reasoning calculation paths in training data can equip the LLMs with finding diverse reasoning logic for solving math problems.

## 5 CONCLUSIONS

In this paper, we are investigating the scaling relationship in supervising math reasoning abilities with large language models. We find the relationship between math performance and pre-training losses, supervised data amount, and distinct reasoning paths. We find that better language models benefit less with SFT and RFT, and the most important thing is to pre-train a better language model towards excellent math reasoning abilities.

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

## A    DETAILED EXPERIMENT SETTING

### A.1    SFT ON GSM8K

We fine-tune GSM8K with 3 epochs and a batch size of 128 on NVIDIA A100 GPUs. We use 8 GPUs for 7B and 13B models, 16 GPUs for 33B models, and 32 GPUs for 65B and 70B models during fine-tuning. We use a peak learning rate of 2e-5 with a $3\%$ learning rate warmup. We evaluate the results on the final epoch. We use greedy decode to calculate maj1@1 and decode with temperature 0.7 to calculate maj1@100.

### A.2    SFT ON DOWNSAMPLED GSM8K

We random downsample GSM8K dataset for fine-tuning. We find that using 3 epochs for little data will result in very poor results which are listed in Table 4. We search training epoch among $\{3, \frac{3}{data\ fraction}\}$ and evaluate the latest epoch. We report better test results among these two different epoch settings.

### A.3    REJECTION SAMPLING FINE-TUNING ON GSM8K

We use an SFT model $\pi$ to sample on training dataset for $k = 100$ times with a temperature of 0.7. We extract the equation list in generated reasoning paths by finding $<<equation>>$ first, removing all white spaces, and joining the equation string list by a special symbol to a string (called it get_equation in our algorithm) for deduplication. We select the reasoning paths by this algorithm:

---
**Algorithm 1:** Reasoning Path Selection

---
**Data:** Reasoning paths for question $q$, $\mathcal{R}_q$
**Result:** Selected reasoning paths for question $q$, $\mathcal{R}_q^s$

1 Initialize selected reasoning paths, $\mathcal{R}_q^s = \text{list}()$
2 Initialize appeared equation set, $\mathcal{E}_q^s = \text{set}()$
3 **for** $r$ in $\mathcal{R}_q$ **do**
4     **if** *get_equation(r)* $\notin \mathcal{E}_q^s$ **then**
5         $\mathcal{R}_q^s$.append($r$);
6         $\mathcal{E}_q^s$.update([*get_equation(r)*])
7     **end**
8     **else**
9         find $r^s \in \mathcal{R}_q^s$ s.t. *get_equation($r^s$)* = *get_equation(r)*;
10         **if** $\sum_{i:r_i^s \in \mathcal{E}_q^s, r_i^s \neq r^s}$ *Levenstein_dist($r$, $r_i^s$)* $> \sum_{i:r_i^s \in \mathcal{E}_q^s, r_i^s \neq r^s}$ *Levenstein_dist($r^s$, $r_i^s$)* **then**
11             $r^s = r$;
12         **end**
13     **end**
14 **end**

---

We are trying to find the most dissimilar reasoning paths based on Levenstein distances. The idea comes from we want diverse reasoning paths for better generalization.

# B   DETAILED RESULTS OF SFT AND RFT

We list detailed results of SFT and RFT in Table 4 and  5.

| Model | Data | Epoch | 7B | 7B-2 | 13B | 13B-2 | 33B | 65B | 70B-2 |
|---|---|---|---|---|---|---|---|---|---|
| ICL-8shot | 0 | 0 | 11.0 | 14.6 | 17.8 | 28.7 | 35.6 | 50.9 | 56.8 |
| SFT | 1/32 | 96 | 9.5 | 10.1 | 8.6 | 17.1 | 18.6 | 25.2 | 27.4 |
| SFT | 1/16 | 48 | 14.3 | 15.5 | 14.2 | 23.9 | 25.9 | 28.9 | 33.6 |
| SFT | 1/8 | 24 | 17.9 | 20.8 | 18.4 | 28.5 | 31.6 | 35.8 | 38.9 |
| SFT | 1/4 | 12 | 21.6 | 27.7 | 26.7 | 36.3 | 38.4 | 45.6 | 46.9 |
| SFT | 1/2 | 6 | 29.0 | 33.1 | 35.2 | 43.7 | 48.6 | 50.5 | 57.5 |
| SFT | 1/32 | 3 | 7.8 | 14.2 | 0.0 | 5.9 | 25.3 | 28.9 | 15.8 |
| SFT | 1/16 | 3 | 12.7 | 16.2 | 7.4 | 27.7 | 29.2 | 39.5 | 52.8 |
| SFT | 1/8 | 3 | 16.5 | 21.8 | 19.5 | 33.4 | 39.3 | 46.0 | 57.8 |
| SFT | 1/4 | 3 | 22.7 | 28.1 | 27.4 | 37.5 | 44.6 | 50.4 | 57.8 |
| SFT | 1/2 | 3 | 30.9 | 34.6 | 36.1 | 45.3 | 50.8 | 55.6 | 61.0 |
| SFT | 7.4K | 3 | 35.9 | 41.6 | 43.0 | 50.0 | 54.6 | 59.3 | 63.2 |
| RFT no dedup | 1/32 | 3 | 37.5 | - | - | - | - | - | - |
| RFT no dedup | 1/16 | 3 | 38.3 | - | - | - | - | - | - |
| RFT no dedup | 1/8 | 3 | 41.1 | - | - | - | - | - | - |
| RFT no dedup | 1/4 | 3 | 41.2 | - | - | - | - | - | - |
| RFT no dedup | 1/2 | 3 | 43.9 | - | - | - | - | - | - |
| RFT no dedup | 400K | 3 | 43.6 | 46.7 | 46.9 | 53.7 | - | - | - |
| RFT k=1 | ~12K | 3 | 37.6 | 43.4 | 42.7 | 52.1 | - | - | - |
| RFT k=3 | ~15K | 3 | 39.0 | 45.3 | 45.2 | 51.9 | - | - | - |
| RFT k=6 | ~18K | 3 | 39.5 | 45.6 | 46.8 | 52.2 | - | - | - |
| RFT k=12 | ~22K | 3 | 41.6 | 45.3 | 48.0 | 53.1 | - | - | - |
| RFT k=25 | ~28K | 3 | 40.9 | 46.5 | 46.0 | 52.6 | - | - | - |
| RFT k=50 | ~35K | 3 | 40.7 | 47.0 | 49.4 | 54.5 | - | - | - |
| RFT k=100 | ~47K | 3 | 41.7 | 47.5 | 49.1 | 54.8 | 54.5 | - | - |
| RFT-U13B | 104K | 3 | 49.3 | 50.3 | 52.1 | 55.4 | 56.5 | 59.0 | 62.3 |
| RFT-U33B | 110K | 3 | 49.1 | 51.2 | 51.4 | 55.3 | 57.9 | 59.7 | 64.8 |

Table 4: Detailed numerical results in this paper, some experiments are still under running. We report maj1@1 (accuracy) in this table.

| Setting | 7B | 7B-2 | 13B | 13B-2 | 33B | 65B | 70B-2 |
|---|---|---|---|---|---|---|---|
| ICL-8shot | 11.0/18.1 | 14.6/- | 17.8/29.3 | 28.7/- | 35.6/53.1 | 50.9/69.7 | 56.8/- |
| SFT | 35.9/48.7 | 41.6/55.4 | 43.0/55.2 | 50.0/61.7 | 54.6/72.6 | 59.3/69.7 | 63.2/73.5 |
| RFT k=100 | 41.7/52.7 | 47.5/58.7 | 49.1/59.9 | 54.8/65.4 | 54.5/- | - | - |
| RFT-U13B | 49.3/61.8 | 50.3/65.6 | 52.1/66.2 | 55.4/69.1 | 56.5/- | 59.0/- | 62.3/- |
| RFT-U33B | 49.1/61.6 | 51.2/64.1 | 51.4/66.3 | 55.3/69.1 | 57.9/71.3 | 59.7/- | 64.8/- |

Table 5: Detailed numerical results in this paper, some experiments are still under running. We report maj1@100 in this table.

# C   CASE STUDY OF RFT

In this section, we present the cases of the training samples from rejection sampling. The case studies would shed light on how RFT potentially improves the mathematical reasoning performance of LLMs. The cases are shown in Table 6. As aforementioned, RFT considers the reasoning paths with different calculation processes regarding equation forms or orders, leading to the correct answers. In the cases from Table 6, all the reasoning paths from RFT result in the correct answer of 10, while the calculation processes of reasoning are diverse. Path 1 and 2, as well as Path 4 and 5, are different in the equation forms as highlighted in red. Path 1 and 2 present a two-step calculation reasoning process while Path 4 and 5 alter to a one-step calculation reasoning process. The case demonstrates that rejection sampling can potentially provide more supervision signals that improve mathematical reasoning performance. The filtered reasoning paths sampled from LLMs themselves are of similar quality to the reasoning demonstrations from human annotations.

| Query A | Weng earns $12 an hour for babysitting. Yesterday, she just did 50 minutes of babysitting. How much did she earn? |
|---|---|
| Path 1 | Weng earned 12/60 = \$⟨⟨12/60=0.2⟩⟩0.2 an hour for the 50 minutes that she worked yesterday. Therefore, she earned \$0.2 x 50 = \$⟨⟨0.2*50=10⟩⟩10 for the 50 minutes that she worked yesterday.#### 10 |
| Path 2 | Weng earns 12 / 60 = \$⟨⟨12/60=0.2⟩⟩0.2 per minute. So, Weng earned 50 * 0.2 = \$⟨⟨50*0.2=10⟩⟩10 for the 50 minutes of babysitting. #### 10 |
| Path 3 | Since there are 60 minutes in an hour, then Weng did 50/60 = ⟨⟨50/60=0.8333⟩⟩0.8333 of an hour of babysitting. So, she earned \$12 x 0.8333 = \$⟨⟨12*0.8333=10⟩⟩10. #### 10 |
| Path 4 | She earns 12 * 50 / 60 = \$⟨⟨12*50/60=10⟩⟩10. #### 10 |
| Path 5 | Weng earns 12 * (50 / 60) = \$⟨⟨12*(50/60)=10⟩⟩10. #### 10 |
| Query B | A local college is offering German lessons and currently has 8 students enrolled. Through advertising, 8 more became interested but a fourth of these dropped out within a day. 2 more got frustrated and left. The class then rallied to show how simple the course actually is and increased enrollment by 5 times the amount of students already enrolled in the class, but 2 had to drop it because of scheduling conflicts. After one last rally, 6 more people enrolled. As the days passed, half of the class eventually dropped, and half of the remaining students graduated. How many are still enrolled? |
| Path 1 | 8+8=⟨⟨8+8=16⟩⟩16 students are enrolled in the German class. 8*.25=⟨⟨8*.25=2⟩⟩2 dropped out from the new recruits. 2 more left, so the class had 16-2-2=⟨⟨16-2-2=12⟩⟩12 students left. There were 12*5+12=⟨⟨12*5+12=72⟩⟩72 total students in the class after the rally. Two people had to drop the class, so there were 72-2=⟨⟨72-2=70⟩⟩70 left. 70+6=⟨⟨70+6=76⟩⟩76 were enrolled after the final rally. 76/2=⟨⟨76/2=38⟩⟩38 are left after half the class dropped. 38/2=⟨⟨38/2=19⟩⟩19 are left after half the class graduated. #### 19 |
| Path 2 | 8+8=⟨⟨8+8=16⟩⟩16 students are enrolled after the first rally. 16/4=⟨⟨16/4=4⟩⟩4 dropped out. 16-4=⟨⟨16-4=12⟩⟩12 remain. 12*5+12=⟨⟨12*5+12=72⟩⟩72 after the second rally. 72-2=⟨⟨72-2=70⟩⟩70 after 2 more dropped. 70+6=⟨⟨70+6=76⟩⟩76 after 6 more enrolled. 76/2=⟨⟨76/2=38⟩⟩38 remain after half the class dropped. 38/2=⟨⟨38/2=19⟩⟩19 remain after half the class graduated. #### 19 |
| Path 3 | Advertising brought in 8+8=⟨⟨8+8=16⟩⟩16 students. Of these, 16/4=⟨⟨16/4=4⟩⟩4 dropped out. 16-4-2=⟨⟨16-4-2=10⟩⟩10 remained. Then 10*5=⟨⟨10*5=50⟩⟩50 more enrolled. This brought the class to 50+10=⟨⟨50+10=60⟩⟩60 students. Then 60+6=⟨⟨60+6=66⟩⟩66 enrolled. So there were 66+10=⟨⟨66+10=76⟩⟩76 students. Then 76/2=⟨⟨76/2=38⟩⟩38 dropped. So 76-38=⟨⟨76-38=38⟩⟩38 remained. Then 38/2=⟨⟨38/2=19⟩⟩19 graduated. So 38-19=⟨⟨38-19=19⟩⟩19 were left. #### 19 |

Table 6: Cases of generated reasoning paths with different reasoning complexity from rejection sampling for RFT. The calculations are highlighted in red.

## D  TOWARDS EXCELSIOR MATHEMATICAL REASONING

From our findings, there are two main factors that can improve mathematical reasoning abilities given a preset amount of human-annotated samples, including: (1) Pre-training the LLMs to lower losses; (2) Augmenting fine-tuning with rejection sampling. Through extensive experiments, we empirically verify the scaling relationships between the mathematical reasoning performance of LLM with both factors respectively. Out of the consideration of sustainable NLP, in this section, we investigate the possible computational resources required to extrapolate the mathematical performance of LLMs by both factors and discuss how to improve the performance more efficiently.

We estimate the pre-training, SFT, RFT inference, and RFT FLOPs following Kaplan et al. (2020) and GPU times in Table 7. We can find that the cost times of SFT ($\sim 1 \times 10^{-5}$) and RFT ($\sim 1 \times 10^{-4}$) are negligible compared to pre-training. One can always use SFT and RFT to improve models' performance. However, it could be hard to use RFT to further boost performance. Since we need much more sampling counts (at an exponential level) to increase distinct reasoning paths and there exists an upper bound of distinct reasoning path amount for a given math reasoning question.

We assume that performance follows RFT>SFT>ICL, from the findings in this paper we know the improvement speed follows RFT<SFT<ICL. And if we have an omnipotent language model which has a pre-training loss that is the same as the corpus randomness, it could have RFT = SFT

| Model size | 7B | 7B-2 | 13B | 13B-2 | 33B | 65B | 70B |
|---|---|---|---|---|---|---|---|
| Pre-train FLOPs | $4.2 \times 10^{22}$ | $8.4 \times 10^{22}$ | $7.8 \times 10^{22}$ | $1.6 \times 10^{23}$ | $2.7 \times 10^{23}$ | $5.5 \times 10^{23}$ | $8.4 \times 10^{23}$ |
| SFT FLOPs | $1.7 \times 10^{17}$ | | $3.3 \times 10^{17}$ | | $7.7 \times 10^{17}$ | $1.3 \times 10^{18}$ | $1.7 \times 10^{18}$ |
| RFT Inference FLOPs | $1.4 \times 10^{18}$ | | $2.6 \times 10^{18}$ | | $6.9 \times 10^{18}$ | $1.4 \times 10^{19}$ | $1.8 \times 10^{19}$ |
| RFT-U33B FLOPs | $3.0 \times 10^{18}$ | | $5.7 \times 10^{18}$ | | $1.3 \times 10^{19}$ | $2.2 \times 10^{19}$ | $3.0 \times 10^{19}$ |
| Pre-train GPU hrs | 82k | 184k | 135k | 368k | 530k | 1022k | 1720k |
| SFT GPU hrs | 0.6 | | 4 | | 40 | 74 | 80 |
| RFT Inference GPU hrs | 10 | | 0.1k | | 0.1k | 4.3k | 4.5k |
| RFT-U33B GPU hrs | 9 | | 62 | | 0.6k | 1k | 1.2k |
| ICL Accuracy | 11.0 | 14.6 | 17.8 | 28.7 | 35.6 | 50.9 | 56.8 |
| SFT Accuracy | 35.9 | 41.6 | 43.0 | 50.0 | 54.6 | 59.3 | 63.2 |
| RFT-U33B Accuracy | 49.1 | 51.2 | 51.4 | 55.3 | 57.9 | 59.7 | 64.8 |

Table 7: The statistics of FLOPs and GPU hours required for pre-training, SFT, RFT inference, and RFT. We take the pre-training GPU hours from Touvron et al. (2023a;b). The GPU hours for RFT inference are calculated for 7,473 train set questions and 100 samples per question. To make the best of GPUs and properly fit models into the GPU memory, we tune the inference batch size. For 33B, 65B, and 70B models, we use DeepSpeed ZeRO3 (Rasley et al., 2020) for distributed training. All the GPU hours are based on NVIDIA A100 80GB GPU. Note we use non-embedding parameters to compute FLOPs in our experiments.

= ICL = 100. Thus when you pre-train a better language model (i.e., smaller pre-training loss), your model's performance still follows RFT>SFT>ICL but their performance gaps are diminishing. Since you can obtain an RFT model without too much effort (compared to pre-training), then the most important thing we should do is to decrease the model's pre-training loss. From LLaMA-7B to LLaMA2-7B, it needs to add $4.2 \times 10^{22}$ FLOPs to obtain a 2.1 improvement in the RFT-U33B setting with a 0.05 pre-training loss decrease. From LLaMA-7B to LLaMA-13B, it adds $3.6 \times 10^{22}$ FLOPs to obtain a 2.3 improvement in the RFT-U33B setting with a 0.07 pre-training loss decrease. While minimizing pre-training loss is expensive compared to SFT and RFT, we believe other abilities may follow a similar pattern and better pre-training can benefit all other tasks.

# E  LIMITATIONS

In this paper, we miss the following parts which are very important for building math reasoning abilities for LLMs and should be discussed in the revised version of this paper or future works.

- Pre-training on the math-related corpus. This is obviously useful shown in Lewkowycz et al. (2022). While the pre-training loss obtained here cannot align with general domain pre-trained models' losses.
- We do not regress any scaling laws in this paper since many numbers are estimated and pre-training losses, ICL prompts and SFT settings of various models may not be aligned.

# F  PRELIMINARY EXPERIMENTS

## F.1  SELF QUERY AUGMENTATION

Through our preliminary experiments and case studies, the errors made by the fine-tuned LLMs are partly attributed to the incorrect reasoning chains where LLMs mistakenly understand the context information or fail to consider all the information in the queries. Although such incorrect reasoning chains lead to wrong answers to the original queries, the reasoning chains themselves represent reasonable logic. For example, for the query *Josh decides to try flipping a house. He buys a house for $80,000 and then puts in $50,000 in repairs. This increased the value of the house by 150%. How much profit did he make?*, a fine-tuned LLaMA model predicts *The value of the house increased by 80,000\*.15=$12,000. So the house was worth 80,000+12,000=$92,000. So he made a profit of 92,000-80,000-50,000=$42,000* where the model erroneously interprets *150%* as *15%*, but the reasoning chain is reasonable if we ignore the error.

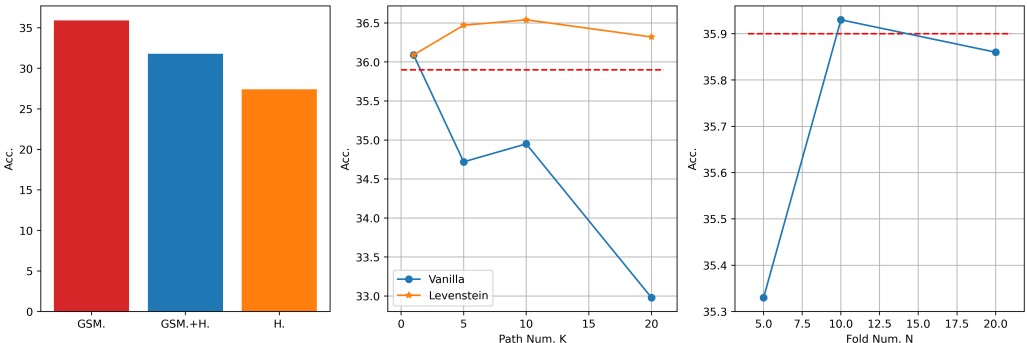

Figure 8: Results for different methods of self data augmentation. GSM. and H. represent GSM8K and Hindsight respectively. The red dotted lines in the middle and right figures represent the results of vanilla fine-tuning on GSM8K.

Therefore, such wrong predictions made by the LLMs may be correct under other queries (if we change *150%* to *15%* in the above example). We conduct experiments to generate queries for the predicted reasoning chains. This is a similar idea to the hindsight experience replay (Andrychowicz et al., 2017) in reinforcement learning where the method is designed to deal with the sparse reward problems by changing the original objectives for the failed samples to form samples with positive rewards. Such an idea was recently adopted by HIR (Zhang et al., 2023) to better align LLMs with instructions.

Concretely, we reformat GSM8K reversely by predicting the query given the corresponding ground-true reasoning result and then we fine-tune a LLaMA model on the reversed task. We use this model to generate queries on the predicted reasoning chains by a normally fine-tuned LLaMA model on the training set of GSM8K, formalizing a training sample for augmentation. We experiment on the LLaMA 7B model and fine-tune models on the data mixing original and generated samples or solely on generated samples.

The results are shown in the left subfigure in Figure 8. We can see that fine-tuning with self query augmentation data leads to the worst results, and the performance of mixing the original data with self query augmented data still falls short of that of the original data. The fine-tuned performance for mathematical reasoning does not benefit from the naive idea of self query augmentation. Through several case studies of generated data, we find that there are two major defects in the generated data. The first one is some reasoning chains themselves are not logically reasonable, for example, there may be some calculation errors in the reasoning chains. The second one is that the generated query may not be suitable for a reasoning chain. The query generation model may still erroneously interpret the information in the reasoning chains. Both defects attribute to a mediocre augmented data quality, hence can be possible reasons for the failure of this data augmentation procedure.

### F.2    SELF REVISING AUGMENTATION

We also explore improving the mathematical reasoning abilities of LLMs through revising augmentation. To equip LLaMA with revising abilities, we generate a revising dataset by first sampling $K$ reasoning paths from a fine-tuned LLaMA model, then concatenating the query with one of the sampled reasoning paths using a template, and finally pairing with the ground-true reasoning path to form a training sample. We use a sampling temperature of 0.7 for generating reasoning paths. During inference, we use the fine-tuned revising model to revise the prediction from the normally fine-tuned model.

The results are shown in the middle subfigure of Figure 8. We can see that with $K = 1$ the revising model improves the final accuracy marginally comparing 36.09% to 35.90%. Surprisingly, as we increase $K$, the performances degrade. The possible defect of the revising model is that generated samples on the training set for revising training suffer from a distribution discrepancy with generated samples on the test set for revising inference. The sampled reasoning paths on the training set may

have a larger lexical similarity to the ground true reasoning paths compared to those on the test set. Therefore we try two different procedures to alleviate such an issue.

1. We use the sampled reasoning path with the largest Levenstein distance out of $K$ sampled paths with respect to the ground true path to form a training sample.

2. We split the train set to $N$ folds, and fine-tune a model on each $N-1$ folds and sampling reasoning path on the left fold.

The results are shown in the middle and right subfigures in Figure 8, we can see that when leveraging Levenstein distance for reasoning path selection, the fine-tuned revising model enjoys a performance boost, harvesting uniformly better performance than the fine-tuning baseline across different $K$'s. The results demonstrate that for the revising performance, the lexical diversity of reasoning paths matters when constructing training samples. However, the revising performance does not benefit from the $N$-fold procedure.

## G  ESTIMATING FLOPs OF SFT AND RFT

We mainly follow the notations of (Kaplan et al., 2020) here.

**Training FLOPs**  For each input sample of length $n_{ctx}$ in GSM8K dataset, we can split it into two parts:

$$n_{ctx} = n_Q + n_R \tag{1}$$

where $n_Q, n_R$ denotes the length of question and generated reasoning path and answers respectively.

$$C_{\text{train}} \approx 6 N n_{ctx} N_s \tag{2}$$

where $N_s$ denotes the numbers of samples.

**Inference FLOPs**  We roughly computed the FLOPs of each token during the forward pass:

$$C_{\text{forward}}(n_{\text{ctx}}) = 2N + 2 n_{\text{layer}} n_{\text{ctx}} d_{\text{model}} \tag{3}$$

To ensure the results were more accurate and reliable, we also took into account the Key-Value (KV) cache during the decoding procedure.

$$KV_{\text{cache}} \approx 4 n_{\text{layer}} d_{\text{model}}^2 \tag{4}$$

Therefore, we obtain the FLOPs per token during the forward pass considering the KV cache.

$$
\begin{align}
C'_{\text{forward}}(n_{ctx}) &= 2N + 2 n_{\text{layer}} n_{ctx} d_{\text{model}} - KV_{\text{cache}} \tag{5} \\
&= 24 n_{\text{layer}} d_{\text{model}}^2 + 2 n_{\text{layer}} n_{ctx} d_{\text{model}} - 4 n_{\text{layer}} d_{\text{model}}^2 \tag{6} \\
&= 20 n_{\text{layer}} d_{\text{model}}^2 + 2 n_{\text{layer}} n_{ctx} d_{\text{model}} \tag{7} \\
&\approx 1.66 N + 2 n_{\text{layer}} n_{ctx} d_{\text{model}} \tag{8}
\end{align}
$$

The total inference FLOPs are computed as follows:

$$C_{\text{total}} = N_s \cdot \left[ n_q C_{\text{forward}}(n_q) + \sum_{i=n_q}^{n_q+n_r} i \cdot C'_{\text{forward}}(i) \right] \tag{9}$$

where $N_s$ denotes the numbers of samples. $n_q, n_r$ denotes the average length (tokens) of the user query and generated response respectively. In GSM8K dataset, $n_q \approx 66$ and $n_r \approx 130$.

# H   ADDITIONAL RESULTS

We use the sampled test set (512 samples, truncated at 2,048 input length) from The Pile (Gao et al., 2020) to calculate pre-train losses among different pre-trained language models including LLaMA (Touvron et al., 2023a), LLaMA2 (Touvron et al., 2023b), and Pythia (Biderman et al., 2023).

To understand the scaling relationship in other math reasoning tasks. We conduct experiments on the MATH (Hendrycks et al., 2021) benchmark with LLaMA and LLaMA2 and show results in Table 8 and Figure 10. We find that (a) The pre-training losses are also negatively correlated to SFT performances; (b) The model performance improves with data amount doubles.

We also conduct experiments with SFT and RFT on Pythia series models and show results in Table 9 and Figure 9. We find that observations from our paper still hold for Pythia including (a) The pre-training losses are negatively linear correlated to SFT performance; (b) The model performance has a log-linear relation versus data amount; (c) RFT improves performances of Pythia series models significantly.

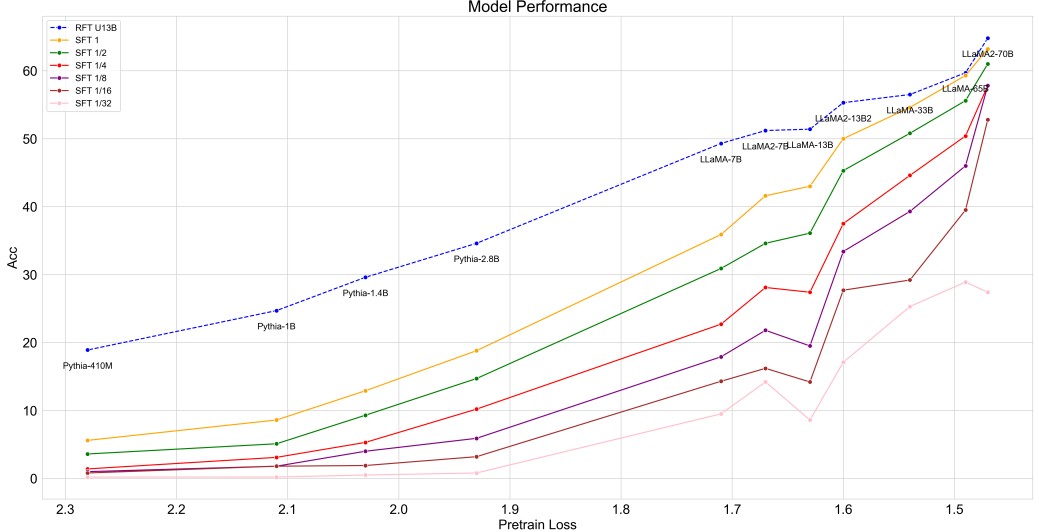

Figure 9: The performance of SFT with different amounts of supervised data on GSM8K using Pythia and LLaMA. Pretrained losses are calculated based on The Pile.

| Setting | 7B | 7B2 | 13B | 13B2 | 33B | 65B | 70B2 |
|---|---|---|---|---|---|---|---|
| Original Pt loss | 1.80 | 1.75 | 1.73 | 1.68 | 1.62 | 1.57 | 1.50 |
| The Pile Pt loss | 1.71 | 1.67 | 1.63 | 1.60 | 1.54 | 1.49 | 1.47 |
| ICL | 2.9 | 2.5 | 3.9 | 3.9 | 7.1 | 10.6 | 13.5 |
| 1/32 | 1.2 | 0.6 | 0.2 | 0.4 | 0.0 | 6.4 | 10.0 |
| 1/16 | 3.0 | 3.8 | 2.4 | 3.6 | 5.0 | 7.2 | 10.0 |
| 1/8 | 2.4 | 3.2 | 2.8 | 3.4 | 6.4 | 8.8 | 10.0 |
| 1/4 | 3.4 | 4.6 | 5.0 | 5.8 | 8.0 | 9.2 | 11.0 |
| 1/2 | 3.8 | 5.0 | 5.2 | 6.2 | 9.4 | 9.8 | 12.6 |
| SFT | 5.4 | 6.0 | 6.6 | 8.8 | 10.2 | 11.6 | 14.6 |

Table 8: MATH accuracy on LLaMA series.

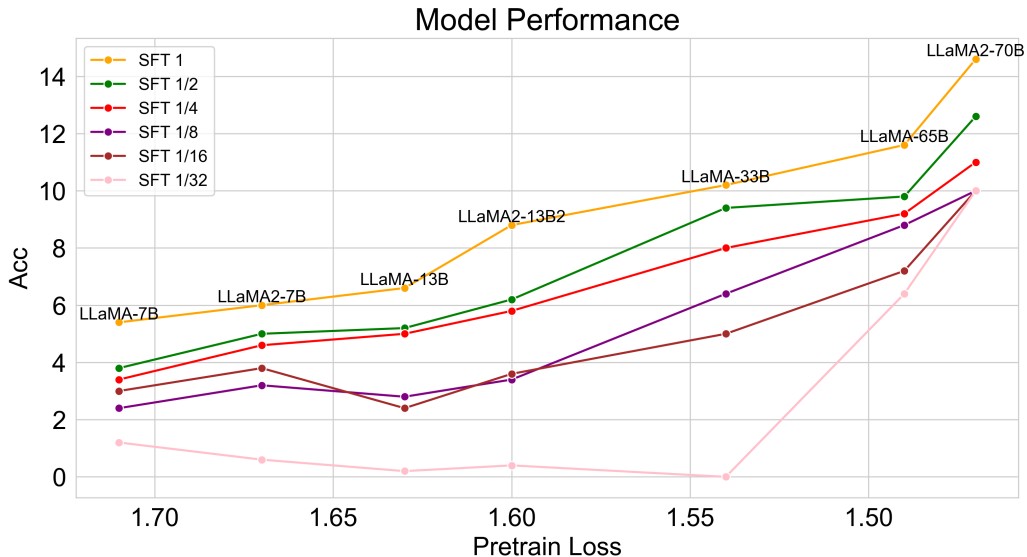

Figure 10: The performance of SFT with different amounts of supervised data on MATH using LLaMA. Pretrained losses are calculated based on The Pile.

| Setting | 410M | 1B | 1.4B | 2.8B |
|---|---|---|---|---|
| The Pile Pt loss | 2.28 | 2.11 | 2.03 | 1.93 |
| 1/32 | 0.2 | 0.2 | 0.5 | 0.8 |
| 1/16 | 0.8 | 1.8 | 1.9 | 3.2 |
| 1/8 | 1.0 | 1.8 | 4.0 | 5.9 |
| 1/4 | 1.4 | 3.1 | 5.3 | 10.2 |
| 1/2 | 3.6 | 5.1 | 9.3 | 14.7 |
| SFT | 5.6 | 8.6 | 12.9 | 18.8 |
| RFT-U13B | 18.9 | 24.7 | 29.6 | 34.6 |

Table 9: GSM8K accuracy on Pythia-v2 series.

