# OpenReview forum: "Scaling Relationship on Learning Mathematical Reasoning with Large Language Models"
_ICLR.cc/2024/Conference — Submitted to ICLR 2024_

### Official Review · Reviewer_J1oN · 2023-11-01

**Soundness:** 2 fair
**Presentation:** 3 good
**Contribution:** 3 good
**Rating:** 5
**Confidence:** 4

**Summary:**

The authors study the effects of pretraining loss, supervised training data amount, and augmented data amount on reasoning performance of tuned large language models. They claim that pretraining loss is negatively linear correlated to the fintuned and in-context learning performance and serves as a more effective performance metric compared to the size of the pretrained model or the amount of pretraining tokens.

A Log-linear trend is reported between training data size and model performance.
The authors also propose to augment the training data using rejection sampling. Essentially, generating multiple answers for each query in the training data and augmenting the dataset using those pairs in which the final answer is correct according to the ground truth. Augmenting the data through rejection sampling from multiple model sizes further improves the reasoning performance of smaller models.

**Strengths:**

The authors did a great job analyzing multiple factors such as, pretraining loss, finetuning data amount and augmented data amount affecting the reasoning performance of the LLM. It shed some light on important factors contributing to making LLMs more proficient in math reasoning.

The introduction and study of filtering the rejection sampling dataset to encourage distinct reasoning path amounts in the augmented data, and its effects on the rejection sampling finetuning performance is an interesting idea.

**Weaknesses:**

The pretraining loss reported are associated with various pretraining datasets and distinct tokenizers, making a direct comparison challenging.

Given that the x-axis in Figure 1 represents the pretraining loss of models of varying sizes, the scaling relationship appears questionable. Wouldn’t it make more sense if the x-axis was the loss of the finetuned models?

The primary concern I have with the study is that the authors do not study different pretraining losses of the same model (identical size and training tokens). Their assertion is that as the pretraining loss decreases, the tuned and ICL performance improve linearly within a certain range. However, it’s important to note that this improvement is not based on the models of the same size or same amount of pretrianing data.

**Questions:**

When combining the rejection sampling data from multiple model sizes do you filter out the generations which do not have a different equation set across model sizes?

What is the value of `k` in Table 3 for `maj1@k` for LLaMA family of models? It would be nice to include it somewhere either in the table or its description.

The claim “as the pretraining loss decreases, the SFT and ICL performance improve linearly within a certain range” appears somewhat unclear and could benefit from additional explanation.

Please refer to the question and concern in the weakness section as well.

---

> ### Author Response · Authors · 2023-11-14
> **General Response**
>
> Thanks for your insightful comment.
>
> General response: Many reviewers are interested in performance for different LLM and different datasets and question our conclusion because of the pre-training losses come from different corpora. We have uploaded a new version with new experiments. We use a random sample of the Pile test corpus to align the pre-train losses of LLaMA, LLaMA2, and Pythia series. We conduct SFT and RFT experiments on GSM8K benchmark with Pythia series. We conduct SFT experiments on MATH benchmark with LLaMA and LLaMA2. Our detailed results are listed in the updated PDF of section H in appendix. I will summary our main findings here.
>
> (a) The pre-training losses are still negatively linear correlated to SFT performance including Pythia series models. (b) RFT improves performances of Pythia series models significantly. (Pythia-410M SFT 5.6 vs RFT 18.9, Pythia-2.8B SFT 18.8 vs RFT 34.6). (c) For MATH benchmark, LLaMA series models still improving performance linearly when double fine-tuning data.
>
> W1: We have calculated the pre-train losses of LLaMA, LLaMA2, and Pythia using the same pretraining dataset - The Pile test set and shown in Appendix H. We find that all our findings still hold.
>
> W2: The fine-tuning losses strongly related to the epoch amount. After the 1st epoch, every sample have been optimized by the model and the fine-tuning losses could be somehow arbitrary to the model performances. Furthermore, the fine-tuning loss (on the test set) is a metric of model performance while the pre-training loss is a performance indicator.
>
> W3: In Appendix H, we conduct the experiment with Pythia-v2. All models from Pythia-v2 series use the same pre-training data and the same amount of optimization steps. We find that Pythia-v2 with RFT still improves performances on GSM8K significantly.
>
> Q1: If all generated samples have the same equation set across model sizes, we will preserve the first one.
>
> Q2: K=100. We will add it to the paper.
>
> Q3: SFT performance and ICL performance is negatively linearly correlated to pre-training losses within some range of pre-training losses. But this cannot hold for all pre-training losses, since pre-training losses is unbounded (0, +\inf), while the model performance is bounded (0, 1). We will rephrase it.

---

### Official Review · Reviewer_zwMw · 2023-11-01

**Soundness:** 3 good
**Presentation:** 3 good
**Contribution:** 2 fair
**Rating:** 6
**Confidence:** 4

**Summary:**

They present an analysis of how performance on the gsm8k dataset improves with models scale / finetuning data. They find that across many models the model's pretraining loss is highly correlated with its performance on gsm8k. They also find that as pretraining loss improves, the benefits from supervised finetuning (SFT) diminish, and models with lower pretraining loss require more finetuning data to surpass the model's in-context-learning performance. Next they study how rejection sampling finetuning (RFT) improves with model scale. The idea is to sample a bunch of outputs from the model and then finetune on the correct samples. They also filter correct outputs that have similar reasoning paths. They find that RFT generally outperforms SFT up to 13B parameters, and benefits less at 33B parameters. Increasing the number of samples for RFT generally improves performance, but less than increasing the size of the finetuning dataset. Lastly they study combining RFT datasets from multiple models, finding that it improves over RFT from a single model up to 13B parameters. Overall by combining RFT data from multiple models they are able to obtain 7B and 13B LMs that are competitive with models that have many more parameters.

**Strengths:**

* Their analysis of the scaling effects of SFT and RFT, in particular the finding that models with lower pretraining loss benefit less from finetuning, are indeed interesting and will contribute to improving our understanding of LM finetuning.
* The question of how LM reasoning improves with model scale is a highly relevant and important question.
* They run a fairly thorough set comparisons and ablations across different finetuning settings, LMs, data amounts. And they do a good job of characterizing how performance on gsm8k changes as each of these parameters are adjusted.

**Weaknesses:**

* Their analysis of how pre-training loss correlates with gsm8k performance is pretty questionable because the loss numbers are for models trained on different datasets and thus are not comparable. While they admit this in the paper, it would be much better if they could make these numbers more comparable. One way to fix this would be to evaluate the loss of all the language models on some standard dataset.
* They only consider gsm8k in this work. It would improve my confidence in their results if they also included other tasks, such as MATH.
* Their RFT method is nearly a special case of STAR [cite], with the only difference being that they also filter reasoning paths for diversity. It therefore is unclear to me to what extent this could be claimed to be a new method.

**Questions:**

* Some of the experiments are shown for up to 70B and others only up to 13B. Would it be possible to present results for all models sizes for all of the different scaling figures?

---

> ### Author Response · Authors · 2023-11-14
> **General Response**
>
> Thanks for your insightful comment.
>
> General response: Many reviewers are interested in performance for different LLM and different datasets and question our conclusion because of the pre-training losses come from different corpora. We have uploaded a new version with new experiments. We use a random sample of the Pile test corpus to align the pre-train losses of LLaMA, LLaMA2, and Pythia series. We conduct SFT and RFT experiments on GSM8K benchmark with Pythia series. We conduct SFT experiments on MATH benchmark with LLaMA and LLaMA2. Our detailed results are listed in the updated PDF of section H in appendix. I will summary our main findings here.
>
> (a) The pre-training losses are still negatively linear correlated to SFT performance including Pythia series models. (b) RFT improves performances of Pythia series models significantly. (Pythia-410M SFT 5.6 vs RFT 18.9, Pythia-2.8B SFT 18.8 vs RFT 34.6). (c) For MATH benchmark, LLaMA series models still improving performance linearly when double fine-tuning data.
>
> W1: We have calculated the pre-train losses of LLaMA, LLaMA2, and Pythia using the same pretraining dataset - The Pile test set and shown in Appendix H. We find that all our findings still hold.
>
> W2: We have added MATH in Appendix H.
>
> W3: We do not mean to propose a new data augmentation algorithm. We want to use RFT to state that augmented fine-tuning performance in math reasoning is based on different reasoning path amount instead of augmented data amount.
>
> Q1: We have conducted SFT, RFT-U13B, and RFT-U33B experiments for all models. We do not conduct RFT (k=100) for models larger than 33B and the reason is two-fold. The first reason is sampling 100 times with 65B/70B models is not cheap (10^19 FLOPs) and the second reason is we have shown that LLaMA-33B SFT models have shown a strong ability for over-fitting (or memorizing) GSM8K problems. LLaMA-33B SFT produces much less different reasoning paths compared to LLaMA-7/13B which is not suitable for RFT.

---

> > ### Comment · Reviewer_zwMw · 2023-11-20
> >
> > Thank you for your response. Adding the additional results on MATH and measuring the log-loss on a held out set is very helpful. I am willing to raise my score to a 6, but it would be great if you could move the new figures in appendix H that use these new logprob calculations into the main paper.

---

> > > ### Author Response · Authors · 2023-11-21
> > >
> > > Thanks for your appreciation. We will update our main paper by considering comments from all reviewers.

---

### Official Review · Reviewer_8Ko8 · 2023-11-05

**Soundness:** 3 good
**Presentation:** 3 good
**Contribution:** 2 fair
**Rating:** 5
**Confidence:** 3

**Summary:**

The paper analyzes the scaling relationship between the math reasoning performance of a supervised fine-tuned LM and pre-training loss, supervised data amount, and augmented data amount. The authors find the pre-training loss is negatively linearly related to the performance while the supervised data amount is log linearly related to the performance. The augmented data amount has a weaker effect than the supervised data amount. All these effects diminish when the LM size increases. The authors also propose a rejection sampling fine-tuning
(RFT) technique, which uses LMs to augment multiple reasoning chains for one training example, to improve the math reasoning performance of an LM. All experiments are performed on the GSM8K dataset.

**Strengths:**

1. The experiments are solid and convincing.
2. The paper is well-written and easy to follow.
3. The proposed RFT significantly improves the performance on GSM8K.

**Weaknesses:**

1. While the experiments are pretty convincing, the final conclusion does not seem to be very surprising or bring new insights. We already know the importance of model size, pre-training loss, and data amount. Since we are not going to design and pre-train a new LM at fine-tuning time, which requires predicting larger LM's performance from smaller ones, I'm not sure why a fine-tuning scaling law would be useful. It just suggests people use the strongest existing pre-training LM, which is exactly what people are doing right now. Maybe the authors can convince me of this during the rebuttal.
2. The proposed RFT feels disconnected from the scaling law experiments. i.e. It does not seem to need to be inspired by the discovered scaling law. RFT basically suggests augmenting the fine-tuning data with LM-generated reasoning paths and rejecting the ones ending with wrong answers. I don't think the idea of data augmentation needs to be inspired by scaling law.
3. Another concern about this paper is that all experiments are limited to one small dataset, GSM8K, which only has 7K+ training examples. Given the scale of the experiments, I can understand this experimental choice. However, I'm still wondering if all the conclusions (e.g. linear with pre-training loss, log-linear with supervised data size) still hold on other math world datasets.

**Questions:**

N/A

---

> ### Author Response · Authors · 2023-11-14
> **General response**
>
> Thanks for your insightful comment.
>
> General response: Many reviewers are interested in performance for different LLM and different datasets and question our conclusion because of the pre-training losses come from different corpora. We have uploaded a new version with new experiments. We use a random sample of the Pile test corpus to align the pre-train losses of LLaMA, LLaMA2, and Pythia series. We conduct SFT and RFT experiments on GSM8K benchmark with Pythia series. We conduct SFT experiments on MATH benchmark with LLaMA and LLaMA2. Our detailed results are listed in the updated PDF of section H in appendix. I will summary our main findings here.
>
> (a) The pre-training losses are still negatively linear correlated to SFT performance including Pythia series models. (b) RFT improves performances of Pythia series models significantly. (Pythia-410M SFT 5.6 vs RFT 18.9, Pythia-2.8B SFT 18.8 vs RFT 34.6). (c) For MATH benchmark, LLaMA series models still improving performance linearly when double fine-tuning data.
>
> W1: To improve a specific ability for LLMs, we have three directions including (1) improve base model via pre-training (2) obtain more human-written fine-tuning data (3) augment more model-generated fine-tuning data. The fine-tuning scaling law (Section 3.2, (2)) and augmented fine-tuning scaling law (Section 3.3, (3)) can help LLM practitioners to decide to generate which kind of data (human-written fine-tuning data or model-generated data). When the amount of SFT data for a task (i.e. gsm8k) is not too much, one can decide to generate human-written data first. When the amount of SFT data is very large or fine-tuning scaling law is similar to augmented scaling law, one should decide to generate data from LLM augmentation.
>
> W2: RFT tries to investigate the augmented data fine-tuning scaling law. Our motivation is not to propose a new algorithm for data augmentation but to understand the augmentation data scaling law. The key finding of RFT is augmented fine-tuning performance in math reasoning is based on different reasoning path amount instead of augmented data amount.
>
> W3: Please see general response.

---

### Official Review · Reviewer_voeM · 2023-11-05

**Soundness:** 3 good
**Presentation:** 3 good
**Contribution:** 2 fair
**Rating:** 5
**Confidence:** 3

**Summary:**

The paper investigates the scaling relationship of factors influencing the mathematical reasoning abilities of large language models (LLMs) through supervised fine-tuning (SFT), in-context learning (ICL), and rejection sampling fine-tuning (RFT) techniques. The authors analyze pre-training losses, the amount of supervised data, and the amount of augmented data and find that pre-training loss has a linear correlation with SFT and ICL accuracy within a certain range. The paper demonstrates that SFT performance has a log-linear relationship with the amount of supervised data, while RFT performance benefits from the increase of distinct reasoning paths. Finally, by combining rejection sampling from multiple models, the authors achieve significant accuracy improvements in multiple LLaMA models.

**Strengths:**

1. The paper is clear, well-organized, and addresses a relevant and important research area.
2. An investigation of how pre-training losses, the amount of supervised data, and the amount of augmented data influencing LLM performance in mathematical reasoning tasks is provided.
3. The paper proposes a new method (RFT) that leverages rejection sampling to generate additional supervised data for improving model performance.

**Weaknesses:**

1. The paper only does evaluation on 1 dataset, limiting its generalizability to other multi-hop reasoning problems.
2. Although the approach is intuitively simple, the amount of computational overhead added makes this approach not appealing.
3. The relationship between downstream task performance and pretraining loss has been well-established since the introduction of BERT. So the results in this paper is not really novel.

**Questions:**

1. Do the authors think the findings of this paper could be generalized and extended to other domains beyond mathematical reasoning tasks? Do the authors expect similar scaling relationships and performance improvements to hold in other LLM applications? Would be nice to see this supported with additional experiments.
2. In the RFT experiments, did the authors observe any instances where the model performance deteriorated or the improvement was insignificant? If so, could you elaborate on such cases and provide possible explanations?
3. Can the authors show similar results with other models than just llama?

---

> ### Author Response · Authors · 2023-11-14
> **General response**
>
> Thanks for your insightful comment.
>
> General response:
> Many reviewers are interested in performance for different LLM and different datasets and question our conclusion because of the pre-training losses come from different corpora. We have uploaded a new version with new experiments. We use a random sample of the Pile test corpus to align the pre-train losses of LLaMA, LLaMA2, and Pythia series. We conduct SFT and RFT experiments on GSM8K benchmark with Pythia series. We conduct SFT experiments on MATH benchmark with LLaMA and LLaMA2. Our detailed results are listed in the updated PDF of section H in appendix. I will summary our main findings here.
>
> (a) The pre-training losses are still negatively linear correlated to SFT performance including Pythia series models.
> (b) RFT improves performances of Pythia series models significantly. (Pythia-410M SFT 5.6 vs RFT 18.9, Pythia-2.8B SFT 18.8 vs RFT 34.6).
> (c) For MATH benchmark, LLaMA series models still improving performance linearly when double fine-tuning data.
>
> W1 & Q3: We have added Pythia on GSM8K and LLaMA on MATH experiments.
>
> W2: In Appendix D, we have covered the RFT inference and training FLOPs, which have a significantly lower computational requirement (1 \times 10^{-4}) compared to pre-training. This level of computational cost is considered acceptable. Although some may argue that using GPT4 API for data augmentation is less expensive, this neglects the computational cost associated with inferencing using GPT4. Additionally, our approach solely relies on models like LLaMA that we aim to enhance, without the need for GPT4. In contrast, other math SFT papers like WizardMATH, Meta-Math, and Mammoth require GPT API for data augmentation.
>
> W3: We are also interested in investigating the relationship between reasoning performance with human-written fine-tuning data amount and model-augmented fine-tuning data amount which is not be researched previously. We also find the distinct reasoning paths are the most important factor for augmentation dates.
>
> Q1: We have conducted experiments on human alignment experiment. We fine-tune LLaMA-7/13/33B models on ShareGPT dataset with (1, 1/4, 1/16, 1/64) amount of data and use MT-Bench to measure the alignment performance. For LLaMA-7B models, we get MT-Bench score with 5.88, 5.85, 5.61, 5.11. For LLaMA-13B models, we get MT-Bench score with 6.13, 6.03, 5.66, 5.24. For LLaMA-33B models, we get MT-Bench score with 6.63, 6.66, 6.17, 5.99. We find that model performances improves with data mount increases in the most scenarios.
>
> Q2: We find that when the pre-trained models get larger (like LLaMA-65B (+0.4) or LLaMA2-70B (+1.6)), the RFT improvement is insignificant. We consider the main reason is larger models have gained considerable reasoning ability during pre-training, and the space for improvement gets smaller for them.

---

### Meta-Review · Area_Chair_GVS3 · 2023-12-14

**Metareview:**

**Paper Summary:**

This paper explores the mathematical reasoning abilities of large language models (LLMs) and how they depend on various factors such as pretraining loss, the amount of supervised data, and the amount of augmented data. The study finds that pretraining loss is a better indicator of model performance than model size and discovers a log-linear relationship between the amount of data and model performance. Additionally, the paper introduces Rejection Sampling Fine-Tuning (RFT) for generating augmented fine-tuning datasets using supervised models. The authors report improvements in model accuracy on the GSM8K dataset through this method.

**Strengths:**

1. Clarity: The paper is well-organized and clear in its presentation (voeM, 8Ko8, zwMw).
2. Solid Experimental Work: The experiments are robust and convincing (8Ko8, zwMw).
3. Strong Results: The RFT method significantly improves performance on GSM8K, especially for small LMs (8Ko8).

**Weaknesses:**

1. Lack of Insights: The relationship between pretraining loss and downstream task performance is not a novel concept, and the findings do not offer new insights (voeM, 8Ko8).
2. Scalability Concerns: As pretrained models increase in size, the improvements from RFT become less significant (General response to voeM).
3. Disconnection and Lack of Novelty: The RFT method does not seem to be directly inspired by the discovered scaling law and appears to be a variation of existing methods (8Ko8, zwMw).

**Decision:**

Based on the reviews, while the paper presents solid experiments and contributes to understanding LM fine-tuning, its limitations in providing new insights and a lack of novelty lead me to not recommend its acceptance.

**Justification For Why Not Higher Score:**

Based on the reviews, while the paper presents solid experiments and contributes to understanding LM fine-tuning, it lacks novelty and doesn't provide new insights.

**Justification For Why Not Lower Score:**

N/A

---

### Decision · Program_Chairs · 2024-01-16

Reject